# MODELLING NEURONAL BEHAVIOUR WITH TIME SERIES REGRESSION: RECURRENT NEURAL NETWORKS ON SYNTHETIC *C. elegans* DATA

## ABSTRACT

Given the inner complexity of the human nervous system, insight into the dynamics of brain activity can be gained from understanding smaller and simpler organisms, such as the nematode *C. elegans*. The behavioural and structural biology of these organisms is well-known, making them prime candidates for benchmarking modelling and simulation techniques. In these complex neuronal collections, classical white-box modelling techniques based on intrinsic structural or behavioural information are either unable to capture the profound nonlinearities of the neuronal response to different stimuli or generate extremely complex models, which are computationally intractable. In this paper we investigate whether it is possible to generate lower complexity black-box models that can capture the system dynamics with low error using only measured or simulated input-output information. We show how the nervous system of *C. elegans* can be modelled and simulated with data-driven models using different neural network architectures. Specifically, we target the use of state of the art recurrent neural networks architectures such as LSTMs and GRUs and compare these architectures in terms of their properties and their RMSE, as well as the complexity of the resulting models. We show that GRU models with a hidden layer size of 4 units are able to accurately reproduce the system's response to very different stimuli.

## 1 INTRODUCTION

The study of the human brain is probably one of the greatest challenges in the field of neuroscience. Recent developments in experimental neuroscience have considerably increased the availability of novel recordings and reconstructions shedding further light into the structure and function of the brain as well as many other systems. But understanding the complexities behind the relations between structure and function as well as the behaviour of such systems across multiple scales in these neuronal collections is constrained by the methods available to study them. This challenge has raised interest in many related fields, such as electrophysiological analysis, imaging techniques, brain-related medicine, computational modelling and simulation, model reduction. Many of these efforts, while not directly providing specific information regarding structural or functional dynamics, do supply large volumes of recordings, measurements or simulations of observable input-output behaviour. The availability of these large datasets raises the question whether low complexity, data-driven, black-box models can be used to model such input-output relations with low error, avoiding the excessive inner detail which may not be known or available.

To determine whether such an approach can be used for large, complex systems, one research direction is the study of smaller and simpler nervous systems, for which the underlying principles of network organization and information processing are easier to postulate. These organisms can become useful models to gain insight into the fundaments of neuronal dynamics and whole brain organization, to validate hypotheses, to develop and test modelling methods, simulation instruments and model reduction techniques. The hope is that the knowledge gained from these analyses and the techniques developed for these simpler organisms can later be used to model more complex systems.

*Caenorhabditis Elegans* (*C. elegans*) belongs to this category of organisms and is quickly becoming one of the benchmarks in whole brain organization studies. *C. elegans* is a nematode (roundworm) of

about 1 mm in length with a compact nervous system consisting of less than 1000 cells across all sexes and around 15000 connections (Cook et al., 2019). This rather small nervous system allows the worm to solve basic problems such as feeding, predator avoidance and mate-finding. The relative simplicity of *C. elegans* allowed for its almost complete description from different perspectives and scales, from its genetics and genomics to the molecular biology, structural anatomy, neuronal function, circuits and behaviour. This information is available in comprehensive databases of genetics and genomics (Hunt-Newbury et al., 2007), electron micrographs and associated data, online books and atlases of the neurobiology, structural and behavioural anatomy (Jackson et al., 2014). Creating a realistic model that encapsulates all this information is not a trivial task. Open-source databases of digitally reconstructed neurons (Gleeson et al., 2019), computational models (Szigeti et al., 2014) and collaborative solutions (Cantarelli et al., 2018) are opening the door for more flexible, multi-scale and multi-algorithm simulation environments for *C. elegans* and other complex biological systems.

The underlying models are based on the connectome, the map of the neuronal connections in the brain. Usually described as a neuronal network, the connectome is a graph where the nodes are the neurons and the edges represent the synapses. The complete connectome of *C. elegans* contains 302 neurons for the adult hermaphrodite (Varshney et al., 2011) and 385 neurons for the male (Cook et al., 2019), but for the latter the respective 3D reconstructions are not yet published. Digital reconstructions for the male are only available for the posterior nervous system of 144 neurons (Jarrell et al., 2012).

The more complex the organism, the more complicated the resulting model, needing more computationally demanding and potentially intractable simulations of its dynamic behaviour. This increased complexity stems from the detailed modelling of the internal structure. However, in many cases, especially of highly complex systems, this detail is not available since the internal mechanisms may not be well known or mapped or it may be simply impossible to examine and record. For that reason, frequently one is really only interested in the peripheral, or input-output behaviour. This motivates our efforts not only to place the focus more on observable input-output data, as well as to try and generate reduced models that avoid extraneous detail not necessary to explain these peripheral relations.

In this work we propose a methodology for generating a reduced order model of the neuronal behaviour of organisms using only peripheral information. We use *C. elegans* as a proxy for our study. Realistic models of *C. elegans*, which take into account spatial distribution and biophysical properties of neuronal compartments have been reported in the literature (Gleeson et al., 2018). We start with a similar model created in-house. Our model comprises the complete connectome of the adult hermaphrodite of *C. elegans*, with 302 multi-compartmental neurons and 6702 synapses (Varshney et al., 2011). The model was validated (Anonymous, 2021) against four scenarios described in related literature (Kim et al., 2019): Forward Crawling Motion (FCM) for the full network, Ablation of AVB interneurons + FCM, Ablation of AVA interneurons + FCM and the Nictation behaviour. We reproduce here the FCM scenario, in which we apply stimulus on the touch sensitive sensory neurons and the interneurons known to be part of the forward movement circuit and we check the activity of the motor neurons associated with forward locomotion. Since we find strong activity in most of these neurons, we conclude that the worm moves forward.

Based on synthetic data extracted from this high-fidelity model, we create a completely equation-free data-driven model assuming no prior knowledge of the original system's structure and equations, using neural networks trained on datasets representing the system's response to different input signals. The ultimate goal is to generate a reduced model to replace the original, detailed one. This reduced model should be able to reproduce with reasonably low RMSE the behaviour of the realistic model while having fewer degrees of freedom. In this work we focus on the issue of reduced RMSE, which we equate to fidelity in reproducing the system dynamics, showing that we can produce sufficiently accurate models for analysing the behaviour of the *C. elegans* nervous system using neural networks.

## 2  RELATED WORK AND CONTEXT

The connectome-based models mentioned above are often termed white-box models, as they are based on direct knowledge and access to the internal structure and parameters' values of the modelled system. These are distributed models, where each neuron has a 3D description and position in space and the synapses are associated with neuronal sections. Such models enable highly accurate simulation of the dynamic behaviour of organisms, but easily become extremely complex as they incorporate detailed structural and functional information of the system.

While the white-box approach ensures access to and evaluation of inner parameters during simulation, it has been shown that the activity of complex networks of neurons can often be described by relatively few distinct patterns, which evolve on low-dimensional subspaces (Karasözen, 2020). This knowledge, together with the ever-present need to avoid potential numerical intractability in large-scale networks with many degrees of freedom, has generated renewed interest in applying model reduction, often also referred to as model compression, to these neuronal networks, including techniques such as Dynamic Mode Decomposition (DMD) (Brunton et al., 2016), Proper Orthogonal Decomposition (POD) (Kellems et al., 2009) and Discrete Empirical Interpolation (DEIM) (Lehtimäki et al., 2019). Depending on the level of morphological accuracy of the underlying models, reduction techniques can have any shade of grey from white-box to black-box, the latter assuming no preliminary knowledge of the system structure and building the model solely out of knowledge of its input-output behaviour.

Black-box approaches are often built upon data-driven models, sometimes learning-based, which have the ability to grasp more naturally and more efficiently the complexity induced by the profound nonlinearities in the neuronal transmission of information. Machine-learning techniques are used to extract data-driven reduced order models for systems arising from differential equations describing the intrinsic dynamics (Regazzoni et al., 2019) and even to extract the governing equations of the estimated model (Sun et al., 2020). It is therefore quite natural to consider using state of the art learning methods for developing reduced models of neuronal behaviour using data obtained from available recordings or even simulations obtained with more complex models.

Especially designed to capture temporal dynamic behaviour, Recurrent Neural Networks (RNNs), in their various architectures such as Long Short-Term Memory (LSTMs) and Gated Recurrent Units (GRUs), have been extensively and successfully used for forecasting or detecting faults in multivariate time series data (Massaoudi et al., 2019), (Gallicchio et al., 2018), (Yuan et al., 2020), (Filonov et al., 2016). Bidirectional LSTMs were used to model genome data by Tavakoli (2019), whereas a combination of CNNs and LSTMs generates a model for epileptic seizure recognition using EEG signal analysis in Xu et al. (2020). An attempt to model the human brain activity based on fMRI using RNNs (LSTMs and GRUs) is reported in Güçlü & van Gerven (2017). In recent years, deep network approaches were used to model realistic neural activity data (Molano-Mazon et al., 2018), (Bellec et al., 2021), (Karampatziakis, 2010). Few studies examined the behavioural output of network models of *C. elegans* using machine-learning techniques. RNNs are generated in a grey-box manner to study the chemotaxis behaviour (Xu et al., 2010) or to predict the synaptic polarities (Lanza et al., 2021) of *C. elegans*, yet these models only include a subset of the connectome.

## 3 METHODS

Given that the starting point is in fact represented by time series data obtained from simulations of the realistic connectome-based model, the modelling task is akin to a sequence to sequence conversion for which the most suitable neural network models are sequential ones.

In this work we analyze the suitability of three of the most commonly used architectures for recurrent neural networks. We start with the least complex unit, the simple RNN, originally proposed in the 1980's to model sequence data (Rumelhart et al., 1986), (Werbos, 1988), (Elman, 1990). The second model used for the recurrent layer is the LSTM unit, developed by Hochreiter & Schmidhuber (1997) and later improved with the introduction of the forget gate to adaptively release internal resources when necessary (Gers et al., 1999). Finally we analyze its sibling, the GRU (Cho et al., 2014).

### 3.1 RECURRENT NEURAL NETWORKS

RNNs (Rumelhart et al., 1986), (Werbos, 1988), (Elman, 1990) are a family of neural networks used for processing sequential data, particularly adept to processing a sequence of values $\mathbf{x}^{(1)}, ..., \mathbf{x}^{(t)}$, and in most cases capable to process sequences of variable length. RNNs appear from the relaxation of the condition on Feedforward Neural Networks (FFNNs) that neurons in a given layer do not form connections among themselves.

Although RNNs, which are trained using Backpropagation Through Time (BPTT) (Werbos, 1990), seem to be a good model for sequential tasks, they are known to suffer mainly from two issues, vanishing and exploding gradients (Bengio et al., 1994). Exploding gradients (Bengio et al., 1994) refer to a large increase in the norm of the gradient during training, which appears due to the explosion

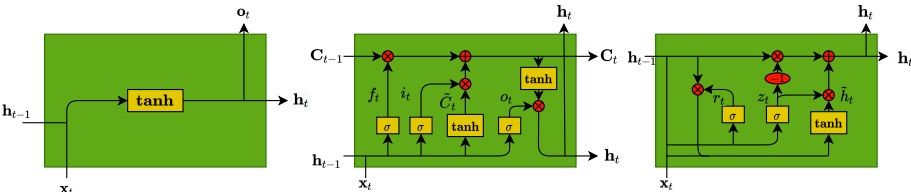

Figure 1: Comparison between the three different units, RNN, LSTM and GRU, respectively.

of long term components that can grow exponentially more than short term ones. This is the less common of the two problems and there are known solutions to handle it, such as the clipping gradient technique (Pascanu et al., 2012). A harder to solve problem is the vanishing gradient issue (Bengio et al., 1994), which refers to when long term components go exponentially fast to norm 0, making it impossible for the model to learn the correlation between temporally distant events.

In order to faithfully reproduce the dynamics of our system the simulations used for generating the datasets require the use of fine time steps, leading to long data sequences. This in turn implies that the response at a given time will depend on values which are far back in the sequence. This situation, however unavoidable, may lead the RNN to experience difficulties in learning our data resulting in a model with unacceptable RMSE.

### 3.2 LONG SHORT-TERM MEMORY

The Long Short-Term Memory unit (Hochreiter & Schmidhuber, 1997) appeared as a solution to the vanishing gradient problem, later improved with the inclusion of the forget gate (Gers et al., 1999). A LSTM unit consists of three main gates, the input gate (1) that controls whether the cell state is updated, the forget gate (2) that defines how the previous memory cell affects the current one and the output gate (3), which controls how the hidden state is updated. Note that LSTM units exhibit a major difference from RNN simple units, since besides the hidden state they also output a cell state to the next LSTM unit, as it can be seen in Figure 1. The LSTM mechanism is described by the following:

$$\mathbf{i_t} = \sigma(\mathbf{W_i x_t} + \mathbf{U_i h_{t-1}} + \mathbf{b_i}) \quad (1) \qquad \mathbf{\tilde{c}_t} = \phi(\mathbf{W_c x_t} + \mathbf{U_c h_{t-1}} + \mathbf{b_c}) \quad (4)$$

$$\mathbf{f_t} = \sigma(\mathbf{W_f x_t} + \mathbf{U_f h_{t-1}} + \mathbf{b_f}) \quad (2) \qquad \mathbf{c_t} = \mathbf{f_t} \circ \mathbf{c_{t-1}} + \mathbf{i_t} \circ \mathbf{\tilde{c}_t} \quad (5)$$

$$\mathbf{o_t} = \sigma(\mathbf{W_o x_t} + \mathbf{U_o h_{t-1}} + \mathbf{b_o}) \quad (3) \qquad \mathbf{h_t} = \mathbf{o_t} \circ \phi(\mathbf{c}_t), \quad (6)$$

where $\mathbf{W_i}, \mathbf{U_i}, \mathbf{W_f}, \mathbf{U_f}, \mathbf{W_o}, \mathbf{U_o}, \mathbf{W_c}$ and $\mathbf{U_c}$, are the weights and $\mathbf{b_i}, \mathbf{b_f}, \mathbf{b_o}$ and $\mathbf{b_c}$ are the biases. All these 12 parameters are usually learned by the model, while $\sigma(\cdots)$ and $\phi(\cdots)$ are the logistic sigmoid and the hyperbolic tangent activation functions, respectively. The outputs of the LSTM unit, the hidden state and the cell state, are computed using (5) and (6), respectively. The computation of the cell state requires the candidate cell state, obtained through (4).

### 3.3 GATED RECURRENT UNITS

The use of LSTM units in recurrent neural networks already produced models that were able to learn very distant dependencies (Gers et al., 2000), but these units are complex structures composed of three gates. For that reason, in 2014 a new type of unit, the GRU (Cho et al., 2014), was suggested, described as follows:

$$\mathbf{z}_t = \sigma(\mathbf{W_z x}_t + \mathbf{U_z h}_{t-1} + \mathbf{b_z}) \quad (7) \qquad \mathbf{\hat{h}_t} = \phi(\mathbf{W_h x_t} + \mathbf{U_h}(\mathbf{r_t} \circ \mathbf{h_{t-1}}) + \mathbf{b_h}) \quad (9)$$

$$\mathbf{r}_t = \sigma(\mathbf{W_r x}_t + \mathbf{U_r h_{t-1}} + \mathbf{b_r}) \quad (8) \qquad \mathbf{h_t} = (\mathbf{1} - \mathbf{z_t}) \circ \mathbf{h_t} + \mathbf{z_t} \circ \mathbf{\hat{h}_t}. \quad (10)$$

In (7), (8) and (9), the weights, $\mathbf{W_z}, \mathbf{U_z}, \mathbf{W_r}, \mathbf{U_r}, \mathbf{W_h}, \mathbf{U_h}$ and the biases $\mathbf{b_z}, \mathbf{b_r}, \mathbf{b_h}$ are the parameters that the model should learn. $\sigma(\cdots)$ and $\phi(\cdots)$ are, again, the logistic sigmoid and the hyperbolic tangent activation functions, respectively.

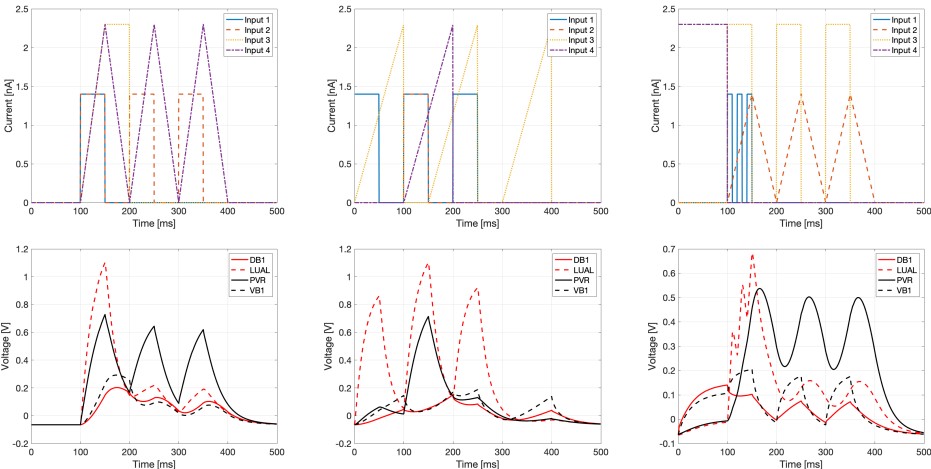

Figure 2: Example input (top row) and output (bottom row) time sequences.

The GRU (Figure 1) is only composed of two gates, the update gate (7) and the reset gate (8). The GRU only outputs the hidden state (10), computed based on the candidate hidden state (9). The update gate controls how much of the past information needs to be passed along to the future, while the reset gate is used to decide how much information the model should forget.

## 4 EXPERIMENTAL SETTING

### 4.1 DATA

The starting model is based on the complete connectome of the adult hermaphrodite of *C. elegans*, with 302 multi-compartmental neurons and 6702 synapses. The neurons are described by 3D geometrical information extracted from NeuroML and LEMS files (Gleeson et al., 2019) for *C.elegans*. We added membrane biophysical properties and connectivity data (chemical synapses and gap junctions) from Gleeson et al. (2018) for the complete connectome. This is a high-fidelity model, since due to the level of detail taken into account we assume it reproduces with fidelity the real output of *C. elegans* neurons. The simulations reproduce the Forward Crawling Motion scenario, by applying varying input currents to two sensory neurons ("PLML", "PLMR") and two interneurons ("AVBL", "AVBR") and record the responses of four neurons known to have strong activity during forward locomotion ("DB1", "LUAL", "PVR" and "VB1"). The resulting system is described in Python and simulated in NEURON (Carnevale & Hines, 2006), one of the traditional neural simulators.

The Python code invokes NEURON to generate the neuronal network, to simulate its behaviour with respect to certain input signals (currents) and to save the responses of the four specific neurons (voltages) previously identified as important for our test case. We simulate the high-fidelity model for 500 ms with two time steps – 0.5 ms and 0.1 ms – and 40 different shapes for the input currents. The input-output variations are extracted into two datasets of 40 snapshot files each, which are further used to train the neural nets. These files are available in the Supplementary material.

To train and tune the hyperparameters, learning rate and batch size, the data was divided into three sets: training, validation and test. The separation of data is done as follows: training set 50% of the data, validation set 25% and for test the remaining 25%. The separation is done by hand, so that validation and test sets are as diverse and demanding as possible. Three examples of the diverse set of inputs and outputs are shown in Figure 2.

### 4.2 MODELLING

The models are developed in Python (Van Rossum & Drake, 2009), using the libraries Keras (Chollet et al., 2015) and Tensorflow (et al., 2015). Details on the code and dependencies to run the experiments are listed in a Readme file available together with the code in the Supplementary material.

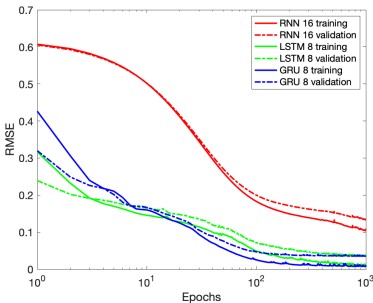 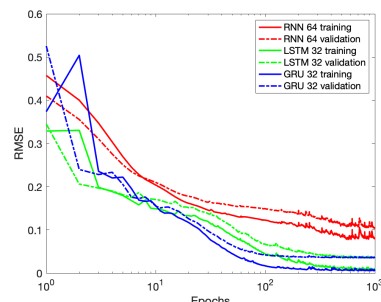

Figure 3: Average training and validation RMSE of ten simulations, with recurrent layers of size 16 and 8 (left) and 64 and 32 (right).

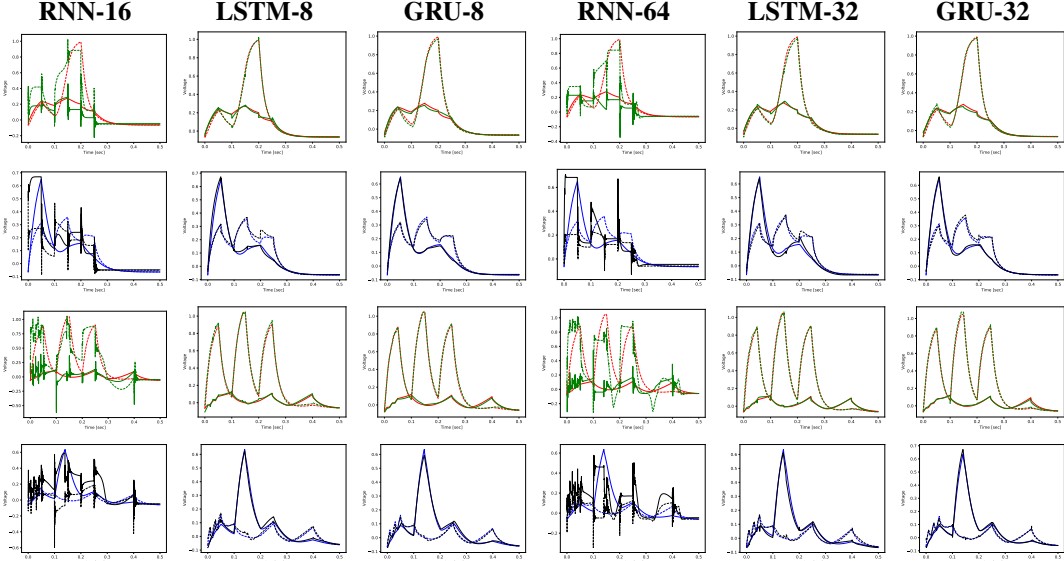

Figure 4: Experiment 1: real (red & blue) and predicted (green & black) sequences for DB1 and LUAL ($1^{st}$ and $3^{rd}$ rows) and PVR and VB1 ($2^{nd}$ and $4^{th}$ rows) for two sequences of the test set (one selected simulation out of ten).

Each model tested consists of one recurrent layer described in Section 3 followed by a dense layer. The dense layer performs a simple linear transformation for each sequence point to convert the output of the recurrent layer, of size "hidden size", into the four outputs.

For a consistent comparison, we fixed the optimizer to Adam (Kingma & Ba, 2015) and the loss function to the root mean squared error. The other hyperparameters, learning rate and batch size, were tuned through experimentation (further details available in the Supplementary material). The experiments on each of the two hyperparameters were conducted separately for the three different

Table 1: The average RMSE of ten simulations, for RNN with 16 and 64 hidden units and for LSTM and GRU with 8 and 32 hidden units, for the iteration with the smallest validation loss.

|  | **RNN-16** | **LSTM-8** | **GRU-8** | **RNN-64** | **LSTM-32** | **GRU-32** |
|---|---|---|---|---|---|---|
| Training | 1.0444e-01 | 9.9371e-03 | 7.7912e-03 | 6.2831e-02 | 7.6058e-03 | 6.7129e-03 |
| Validation | 1.2986e-01 | 3.6109e-02 | 3.4760e-02 | 8.8764e-02 | 3.6437e-02 | 3.5098e-02 |
| Test | 1.3406e-01 | 1.4904e-02 | 1.0005e-02 | 9.5476e-02 | 1.5230e-02 | 1.2384e-02 |

Table 2: The average RMSE of ten simulations obtained with the GRU model, for different sizes of the recurrent layer.

|          | 2 Units    | 4 Units    | 8 Units    | 16 Units   | 32 Units   | 64 Units   |
|----------|------------|------------|------------|------------|------------|------------|
| Training | 4.2999e-02 | 1.0525e-02 | 7.7912e-03 | 6.7886e-03 | 6.7128e-03 | 5.8068e-03 |
| Validation | 5.3732e-02 | 3.4800e-02 | 3.4760e-02 | 3.4961e-02 | 3.5098e-02 | 3.6230e-02 |
| Test     | 6.0047e-02 | 1.1771e-02 | 1.0005e-02 | 9.3625e-03 | 1.2384e-02 | 1.6841e-02 |

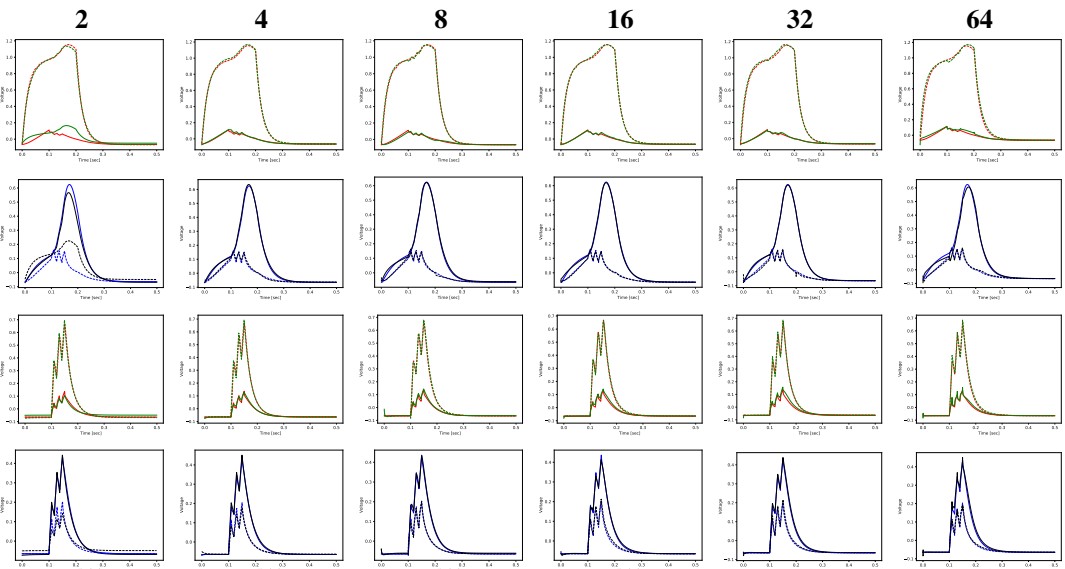

Figure 5: Experiment 2: real (red & blue) and predicted (green & black) sequences for DB1 and LUAL ($1^{st}$ and $3^{rd}$ rows) and PVR and VB1 ($2^{nd}$ and $4^{th}$ rows) for two sequences of the test set (one selected simulation out of ten).

recurrent layers with a fixed hidden size of 16 units, and the other hyperparameter fixed. Each model was trained for 1000 epochs, with the final model chosen as the best iteration on the validation set. After this experimentation we chose to fix the batch size at 32 for all three models, the learning rate at 0.001 for the RNN and at 0.05 for the LSTM and the GRU as these values seemed to provide the best results. Different activation functions were also experimented, but did not end up having a positive impact on the results, therefore we kept the default ones.

## 5 EXPERIMENTS AND RESULTS

In Experiment 1 (Section 5.1) we compare the performance of the three types of layers, RNN, LSTM and GRU. Experiment 2 (Section 5.2) carries a comparison between different sizes of the GRU recurrent layer in order to determine the optimal size under some RMSE constraints. Finally, Experiment 3 (Section 5.3) is an investigation upon the ability of the models to reproduce data resulting from simulations with a finer time step, therefore involving longer sequences with more data points. For all the experiments, the loss is computed as the average RMSE of ten simulations.

### 5.1 EXPERIMENT 1

In this experiment we compare the performances of the three types of units on the dataset corresponding to the coarser time step (0.5 ms). In the interest of fairness we use layers with comparable number of parameters: a RNN with 16 hidden units (404 total parameters) against a LSTM and a GRU, both with 8 units (452 and 348 parameters, respectively); and a RNN with 64 units (4676 parameters) against a LSTM and a GRU with 32 units (4868 and 3684 parameters, respectively).

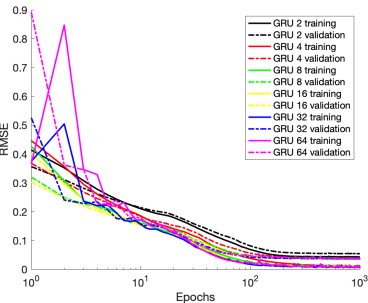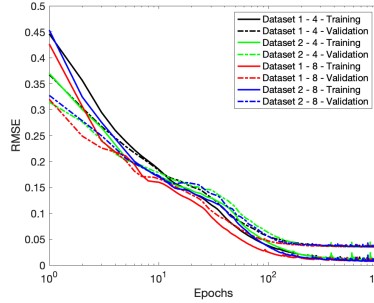

Figure 6: Average training and validation RMSE of ten simulations. Left: for 6 different hidden sizes of the GRU-based recurrent layer, for the iteration with the smallest validation loss. Right: for the two datasets with two different hidden sizes with a GRU-based recurrent layer.

Table 3: The average RMSE of ten simulations obtained with the GRU model, for the two datasets and two sizes of the recurrent layer (4 and 8 hidden units).

|            | Dataset1 - 4 | Dataset2 - 4 | Dataset1 - 8 | Dataset2 - 8 |
|------------|--------------|--------------|--------------|--------------|
| Training   | 1.0525e-02   | 1.0878e-02   | 7.7912e-03   | 8.1769e-03   |
| Validation | 3.4800e-02   | 3.4554e-02   | 3.4760e-02   | 3.5440e-02   |
| Test       | 1.1771e-02   | 1.2107e-02   | 1.0005e-02   | 1.0434e-02   |

Figure 3 shows the evolution of the training and validation losses during the training process. The simple RNN unit tends to take more time to learn, being also slightly less stable during the end of the training process. Although this is not a good indicator, it is not as alarming as the behaviour shown in Figure 4, where it is clear that the simple RNN unit is not able to reproduce the outputs with the desired reduced RMSE, while the LSTM and the GRU perform well. A summary of this experiment's results is shown in Table 1. Since the simple RNN unit did not perform sufficiently well by not being able to reproduce the output with minimal RMSE, we are left with the LSTMs and GRUs units. Given that the GRU is the less complex unit of the two, we consider it the main option and keep the LSTM as an alternative architecture.

## 5.2 EXPERIMENT 2

The GRU, due to its low RMSE and relative simplicity, therefore emerged as the prime candidate unit for our modelling purposes. However, we now want to determine how small the models can be without compromising the overall error. The focus of this second experiment is therefore to test different sizes of the recurrent layer and determine the smallest size that is still able to generate a model with sufficiently low RMSE. We test both the LSTM and GRU units using the dataset with the coarser time step of $0.5$ ms. Since the LSTM does not produce noticeable improvement over the GRU with a similar number of parameters, we only report here the results obtained with the GRU for six different sizes of the recurrent layer: $2, 4, 8, 16, 32, 64$ (Table 2). Figure 6 (left) illustrates the evolution of the training and validation losses during the learning process, where one can see that even for size 8 the model reaches a low and stable loss. In fact, from Figure 5 and Table 2 we can state that a GRU with a size of 4 hidden units is optimal to reproduce the outputs with low RMSE.

## 5.3 EXPERIMENT 3

In this experiment we explore the models' behaviour in the same simulation interval using data sampled with different time steps as this leads to sequences of varying lengths. From a methodology standpoint this is important since even though time-wise the dynamics do not change, the temporal dependencies that the model has to learn are farther back in the sequence, which increases the difficulty of the learning process. We run the model for two datasets, one with the coarser ($0.5$ ms) time step and one with a finer ($0.1$ ms) one. The experiment is done only for the GRU, with 4

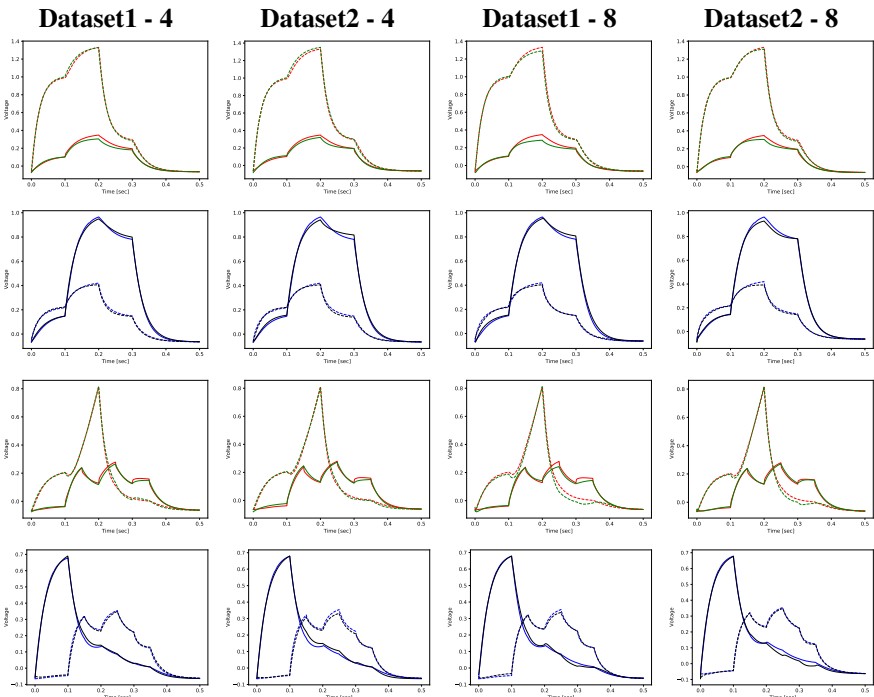

Figure 7: Experiment 3: real (red & blue) and predicted (green & black) sequences for DB1 and LUAL ($1^{st}$ and $3^{rd}$ rows) and PVR and VB1 ($2^{nd}$ and $4^{th}$ rows) for two sequences of the test set (one selected simulation out of ten).

and 8 units. Even though the model takes more time to converge for the finer time step, it ends up stabilizing with a loss of the same order in both cases and the model fits the test data well, as shown in Figure 6 (right) and Table 3. The plots in Figure 7 further strengthen this idea. We can conclude that a GRU with 4 hidden units is consistently able to reproduce the outputs of the original model with a reasonably low RMSE, for various inputs, for data sampled with coarser and finer time steps.

## 6   CONCLUSION

In this paper we create models for the *C. elegans* nervous system with three different recurrent neural networks architectures: simple RNNs, LSTMs and GRUs. The objective is to further generate a low-order description to replace the original, detailed model in the NEURON simulator. To achieve this goal we seek a model as simple as possible and therefore the ideal unit would appear to be the simple RNN. However, this unit does not perform sufficiently well compared to the other two architectures. The LSTM and GRU give comparable results in terms of overall fidelity, measured through RMSE, for different sizes of the recurrent layer. Due to its simplicity, GRU is preferable and with a hidden size of 4 units, is able to reproduce with high fidelity, i.e. low RMSE, the original model's responses to different types of stimuli. Furthermore, from a computational standpoint, explicitly inferring the response of the GRU model to such stimuli will vastly outperform the cost associated with simulating the high-fidelity model within NEURON, which has to solve the set of nonlinear equations implicit in the connectome network. However, to quantify the potential advantage and to provide a fairer comparison would require including our model into NEURON and perform an identical simulation.

Further work will concentrate on improving the automation in choosing appropriate stimuli for the training, validation and test sets as well as optimal parameter selection. This will require a systematic analysis of compression possibilities of the learning-based models with error control. These results nonetheless show that it is feasible to develop recurrent neural network models able to infer input-output behaviours of real biological systems, enabling researchers to advance their understanding of these systems even in the absence of detailed level of connectivity.

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

## SUPPLEMENTARY MATERIAL

The Supplementary Material containing the datasets, the models, the source code and the instructions to run them is available at the following anonymous GitHub repository:

https://anonymous.4open.science/r/ICLR-RNN-CElegans-D2BB

