# OpenReview forum: "Modelling neuronal behaviour with time series regression: Recurrent Neural Networks on synthetic C. elegans data"
_ICLR.cc/2022/Conference — ICLR 2022 Submitted_

### Official Review · Reviewer_okcY · 2021-11-01

**Correctness:** 4
**Technical Novelty And Significance:** 2
**Empirical Novelty And Significance:** 3
**Recommendation:** 8
**Confidence:** 4

**Main Review:**

## Strenghts
- The paper is very well written and is an enjoyable read. It explains
  well the problem it is trying to address and why it's important, and
  it introduces all concepts and techniques it uses in an accessible
  way. I think this is particularly important for this paper, as its
  main audience would presumably be computational neuroscientists, so
  it seems a good idea not to take for granted a deep knowledge of
  (say) recurrent network architectures.
- Addressing the problem of model reduction using off-the-shelf
  machine learning approaches is a timely choice, given the very rapid
  increase of experimental data available for modeling in this field.
- The main thesis of the paper is well supported by the evidence,
  which is laid out in a logical and clear progression.

## Weaknesses
- Mirroring what I wrote above in one of the "strenghts" points: the
  fact that paper is a rather simple application of off-the-shelf RNN
  architectures to a well-defined modeling problem means that the
  paper is not hugely ambitious, and that it doesn't bring much in the
  way of theoretical or conceptual insight; however, in my opinion
  this does not detract from the quality of the work.

## Minor suggestions:
- When discussing related literature, I invite the authors to consider
  the inclusion of a broader selection of papers that in recent years
  applied deep network approaches to modeling neural activity data, as
  the works currently cited in the paper investigate only fMRI data
  (which is only distantly related to actual neural activity), or are
  specific to C Elegans. Two examples that come to mind are
  Molano-Mazon et al, ICLR 2018, *Synthesizing realistic neural
  population activity patterns using generative adversarial networks*,
  and Bellec et al, NeurIPS 2021, *Fitting summary statistics of neural
  data with a differentiable spiking network simulator*.
- If possible, it would be great to have an actual comparison of the
  compute time/cost incurred in running the trained RNN (even just for
  the final selected version of the architecture, the 4- or 8-units
  GRU), versus running the ground-truth simulation with NEURON. I
  apologise if this is given somewhere and I missed it.
- Please make all figure labels bigger. Axes labels, tick labels and
  legend contents are currently almost impossible to read!
- on page 7, last line of section 5.1: "*[...] we consider it the main
  option and keepING the LSTM as an alternative architecture*" → should
  be "keep".



**Summary Of The Paper:**

This paper investigates the use of recurrent neural networks as a
model reduction tool in computational neuroscience. More specifically,
the authors consider the problem of predicting the activity of a set
of four neurons in the C Elegans nematode worm, resulting from the
(simulated) electrical stimulation of other neurons in the animal's
connectome. Three experiments are carried out, testing different
network architectures, network sizes, and the effect of increasing the
temporal resolution of the data. The results show that a small
GRU-based network is sufficient for achieving excellent agreement with
the starting data, which was generated using computationally demanding
simulations of a network of multi-compartimental neurons. The paper
also includes an introduction on popular RNN architectures, and on the
problem of model reduction in computational neuroscience.


**Summary Of The Review:**

This is a solid paper that addresses a question which will be of
interest to part of the ICLR community. The research is somewhat
limited in scope, but very well executed and written up, and overall a
worthy contribution. It is also a good demonstration of the usefulness
of open databases of neuroscience models and data.

My recommendation is to **accept**.

---

> ### Author Response · Authors · 2021-11-18
> **Response from authors of Paper2512 to Reviewer okcY**
>
> Thank you for your suggestions and for the encouraging and positive comments. While we agree that the RNN architectures used are fairly standard, the main goal of the work is indeed to show that black-box techniques using general machine-learning structures can produce accurate models without assumptions on the intrinsic structure of the system under analysis. Most of the existing modelling approaches tend to be application specific, assuming knowledge of internal structure to a varying level of detail and are therefore hard to extend to other systems or organisms.
>
> More work is needed to include our black-box model into NEURON and perform further simulations to ensure a fair comparison with the high-fidelity model. This will require some effort given the complexity and specificity of the NEURON simulator.
>
> We agree that the figures’ labels are not very visible, especially the ones containing real and predicted sequences, this was due to the space limitations. However, we will do our best to improve the visibility of all labels.
>
> All your other suggestions are welcome as well, we will take them into consideration for the final version of the manuscript.

---

> > ### Comment · Reviewer_okcY · 2021-11-18
> > **The author's response mostly addresses the issues I raised**
> >
> > Thank you for your answer, which addresses most of the points I made.
> >
> > I only have one (minor) thing left on which I'm somewhat confused: I'm not sure why you talk about "includ[ing] our black-box model into NEURON". Was this meant to address my request for a computational cost comparison of the ground truth vs the reduced model? If so, I don't understand why you would need to include your model inside NEURON, or even what "including a model inside NEURON" actually means in this context. The way I saw it, you had two simulations for the activity of the same neurons under the same conditions, and you could have just reported how long it took you to run such simulations. But perhaps I'm missing something?

---

> > > ### Author Response · Authors · 2021-11-22
> > > **Response from authors of Paper2512 to Reviewer okcY**
> > >
> > > It appears we might have indeed misunderstood your comment.  We did assume that your point about computational cost referred to a fair comparison of the original, high-fidelity model versus our black-box model.  The only fair comparison in terms of computational cost of model evaluation would need to be conducted on the same machine in the same environment, likely for a longer time period.
> > >
> > > As is, the high-fidelity model is computed inside NEURON, which is a complex system, with a complicated, albeit efficient, timestep control mechanism, that for a given the network structure it builds and solves a set of nonlinear equations at each time point. On the other hand our inference results were obtained on a completely different machine, using a Python script where the equations are directly solved, explicitly, at each time point.  As such the comparisons are not really fair towards the full model. Our inference computation is almost instantaneous for all the stimuli attempted as may be expected of a simple single-layer 4-neuron model.
> > >
> > > In order to include our model into NEURON, we will need to extend NEURON to be able to include custom models, read in the relevant structure (number of layers, number of neurons, type of neurons), coefficients, etc, and will have to ensure that the inference is computed at the right place in the simulation process.

---

> > > > ### Comment · Reviewer_okcY · 2021-11-22
> > > > **Thanks for the clarification**
> > > >
> > > > I see. Overall I think I now understand where you are coming from, and I agree that making a fair comparison could be harder than I thought. However I was not really thinking of something very formal or sophisticated: it may be obvious to you (and to me also, that's kind of why I asked) that the reduced model will be much faster than the NEURON model to simulate. But this may not be the case for all readers, who may only have an intuition about how long it takes to run a small rnn, or to run a simulation with hundreds of detailed neurons in NEURON, but not both. So perhaps there could still be some value in saying something like "as expected from a reduced model, producing simulations with out rnn was much faster than with the original high-fidelity model (insert rough orders of magnitude here). This however is not a completely fair comparison to the NEURON simulator because it was run on a different machine, and because NEURON used an adaptive integration time step. (add any additional reason)"? Just an idea that in my view could be helpful, of course feel free to ignore it.
> > > >
> > > > Having said this, I'll also observe the following, more in detail.
> > > > 1. the issue of running on different machine is not a big deal in my view, as long as this is well explained to the reader. These are probably two different workstation-class computers anyway, I suppose (unless you're running the NEURON model in parallel on a cluster, but I'm not sure how much could a c elegans model benefit from parallelization)? Again I'm not really thinking about providing formal comparisons, but about giving a rough idea of the orders of magnitude involved, in practice, on typical computers.
> > > >
> > > > 2. I hadn't thought about the variable timestep issue. On the one hand, one could always run neuron in fixed timestep mode (I'm not suggesting you do this now). On the other hand, NEURON can do variable timestep integration because what it is integrating are differential equations that are in principle defined for continuous time, and then discretized; but the rnn is an intrinsically discrete-time device. So "including the model in NEURON" really would mean representing the (trained) RNN as a set of differential equations in a sensible way, which is of course a very challenging prospect (sounds like the type of thing that people may have studied already, but I have no references off the top of my head), and entirely outside the scope of this work.
> > > >
> > > > Food for thought. Thank you again for your comments!

---

> > > > > ### Author Response · Authors · 2021-11-22
> > > > > **Response from authors of Paper2512 to Reviewer okcY**
> > > > >
> > > > > Thank you for your quick reply and clarification. We have added a statement to this effect in the Conclusion section, but also tried to avoid confusing the reader with details that would fall outside the scope of this work, such as including a discussion on timestep control.
> > > > >
> > > > > Thank you again and best regards.

---

### Official Review · Reviewer_mRji · 2021-11-02

**Correctness:** 3
**Technical Novelty And Significance:** 1
**Empirical Novelty And Significance:** 2
**Recommendation:** 3
**Confidence:** 5

**Main Review:**


The paper uses a connectome-based in-house model of the worm, utilizing compartmental neuron models and chemical and electrical synapses to generate the ground truth data. In general, I found that insufficient details are provided about how this model was built & validated (how were the biophysical properties chosen, what synapse models were used, etc). The authors state that the simulator is "high-fidelity" because they "assume it reproduces with fidelity the real output of C. elegans neurons.". The model might be high-complexity, but as far as I could see, fidelity was not demonstrated anywhere. In fact, the only test case, also used for all subsequent modeling work with neural networks, is "forward locomotion". In it, 2 sensory neurons and 2 interneurons are stimulated to generate activity in 4 neurons known to be active in locomotion. It's possible that additional information is available in the supplementary materials which I was not able to access ("ERRORS.The value of "offset" is out of range. It must be >= 0 && <= 17825792. Received 17825795"). Even if that's the case, more of this information should be in the main text.

I found section 3.1 with detailed discussion of GRU, LSTM, and problems related to training RNNs a bit too verbose. These are standard and well-known architectures at this point, and could just be briefly introduced with references to the original papers. The reader would benefit much more from learning the details of the C. elegans simulation.

The paper does a good job of systematically testing various variants of recurrent neural networks (LSTM, GRU, vanilla RNN) in a simple setup involving a single recurrent layer followed by a dense layer with 4 outputs. The authors focus on GRU, and conclude that a tiny neural network with just 4 GRU units is sufficient.

That a GRU network works for modeling time-series data is not particularly surprising. I found the experimental work presented in the paper far too limited to yield useful insights into using RNNs as a black-box model of real brain activity. Ideally, the experiments should cover more neurons and behaviors, and variations in synapse strengths (see doi:10.1038/s41586-021-03778-8) and the biophysical parameters. What is missing is also a comparison of the computational complexity of the simulation and the proposed reduced order model. One could also ask whether the model used in the simulation is actually necessary to produce this "forward motion" scenario -- could some neurons be excluded, or simpler neuron models used instead?

Specific comments:
* The text mentions that the separation of the simulation results into train/test/validation was done manually. Please provide more details on the criteria you used to ensure diversity within each group.
* In experiment 3, you state that "[..] data sampled with different time steps as this leads to longer sequences". This should probably be rephrased to highlight that the same physical time is simulated.
* What exactly is the input to the network? Only the 2 currents on the sensory neurons or also on the interneurons?
* Why is it necessary to provide stimulus on interneurons instead of just sensory neurons?




**Summary Of The Paper:**

The paper shows that a small recurrent neural network can be used to predict the activity of 4 neurons in a C. elegans simulation with good accuracy as measured by RMSE.


**Summary Of The Review:**

The paper effectively shows that a tiny GRU network can reproduce a few time-series generated from a larger simulation. The general research direction of building reduced-order models for such simulations is interesting, but the very limited empirical results presented here and lack of details about the simulation make it impossible for me to recommend acceptance.

---

> ### Author Response · Authors · 2021-11-18
> **Response from authors of Paper2512 to Reviewer mRji**
>
> We agree that the validation of the underlying model, termed high-fidelity, is not described in detail in the text as we felt it was not the purpose of this paper. For space reasons we decided to not extend the introduction too much with these details.
> The connectome of C.elegans and the c302 model we reproduced is well-known and already established as the primary benchmark for the realistic behavioural modelling of this organism.
> The precise definition and validation of this C.elegans high-fidelity model was presented in another paper of ours, previously published, which we did not cite to ensure anonymity in the review process. We intend to add that reference to the final paper.
> The model was validated against four scenarios described in related literature. (e.g. J. Kim, W. Leahy, E. Shlizerman. Neural interactome: Interactive simulation of a neuronal system. Frontiers in computational neuroscience, 13:8, 2019):
> Forward Crawling Motion (FCM) for the full network, Ablation of AVB interneurons + FCM, Ablation of AVA interneurons + FCM and the Nictation scenario.
> Also thank you for pointing out the error in accessing the Supplementary material, we have now solved the issue and it should be accessible at the same link.
>
> Regarding your comment about the verbosity of section 3.1, we considered that part of the audience would not have a deep knowledge of recurrent network architectures, so it would be a good idea to briefly introduce the concepts and techniques in an accessible way. We believe this makes the paper more accessible to audiences with varying backgrounds, more neuroscience or more machine-learning inclined.
>
> We thank you for bringing up the mentioned reference. We will take it into consideration. We agree that more work is needed to include our black-box model into NEURON and perform further simulations to ensure a fair comparison with the high-fidelity model. This will require some effort given the complexity and specificity of NEURON.
>
> The majority of the neurons in the connectome are involved in the Forward Motion scenario. A proper validation is done by simulating the full network, then simulating the network without neurons known not to be involved in the forward locomotion circuit (e.g. the AVA interneurons do not influence forward movement), then repeat the simulation without neurons whose presence is important in this scenario (for example the lack of AVB interneurons impedes forward motion). We validated the behaviour of our model in these scenarios and in the Nictation scenario.
> We hope that in the final version, the addition of a short sentence and an accompanying reference regarding the validation of the high-fidelity model will help to clarify this point.
>
> The separation of datasets into train/test/validation is a relevant point that was also brought up by another reviewer. We agree that automating the process of dividing the existing data into training, validation and test sets is important to ensure stimuli and response diversity. While this task was performed manually from our dataset of 40 snapshot files, automation will be required for larger datasets. The criteria chosen was to ensure that the test set included responses that covered the full range, as well as responses with varying time constants.
>
> Thank you for the suggestion about the description of Experiment 3, we will complete the sentence with "data sampled with different time steps as this leads to longer sequences, for the same time range of simulation".
>
> To answer your question about the input of the network, this is composed of 2 currents to stimulate the sensory neurons and 2 currents to stimulate the interneurons, so 4 inputs in total. This is specified in subsection 4.1 Data, as:
> “The simulations reproduce the Forward Motion scenario, by applying varying input currents to two sensory neurons (”PLML”, ”PLMR”) and two interneurons (”AVBL”, ”AVBR”) and record the responses [...]”
>
> The PLM sensory neurons (PLML/PLMR) in C. elegans nervous system are known as posterior mechanoreceptors, they further excite neurons associated with forward crawling motion (see Kim et al., 2019, Chalfie et al., 1985).
> The AVB interneurons (AVBL/AVBR) are also known as driver cells for forward movement and are excited by some internal mechanism of the worm (Kim et al., 2019). It is known that the removal of the AVB interneurons from the network impedes forward locomotion. The neurons chosen to be stimulated, as well as the amplitude of the stimuli are therefore in concordance with the related literature and are designed in such a way so that they allow a proper assessment of the worm’s behaviour in a certain scenario.
>
> To conclude, we hope to have addressed most of the relevant points that you raised in your review and furthermore that the addition of a reference that describes the initial model may lessen some of your concerns. Additionally, we fixed the access to the data, which now allows you to examine the entire dataset used.

---

> > ### Comment · Reviewer_mRji · 2021-11-29
> > **Thanks**
> >
> > Thank you for the detailed explanation. While I agree that it is a good idea to make the paper accessible to a wide audience, statistically speaking, at this point there many more researchers familiar with LSTMs and GRUs than with the details of C. elegans simulations. This is even more so when you consider the audience of ICLR specifically.
> >
> > As I suggested in the review, I think building ML-based reduced order models is an interesting area of work. In order for the paper to be relevant to the wider ICLR audience, I suggest looking into significantly expanding the empirical work and trying to arrive at some general conclusions that would be more broadly applicable beyond the specific case you tested in the present version.

---

### Official Review · Reviewer_Y3zd · 2021-11-08

**Correctness:** 4
**Technical Novelty And Significance:** 3
**Empirical Novelty And Significance:** 3
**Recommendation:** 6
**Confidence:** 5

**Main Review:**

In this paper authors create models for the C. elegans nervous system with three different recurrent neural networks architectures: simple RNNs, LSTMs and GRUs. The objective is to further generate a low-order description to replace the original, detailed model in the NEURON simulator.

Authors should improve the automation in choosing appropriate stimuli for the training, validation and test sets as well as optimal parameter selection and perform a systematic analysis of compression possibilities of the learning-based models with error control.

**Summary Of The Paper:**

Authors show how the nervous system of C. elegans can be modelled and simulated with data-driven models using different neural network architectures. Specifically, they target the use of state of the art recurrent neural networks architectures such as LSTMs and GRUs and compare these architectures in terms of their properties and their RMSE, as well as the complexity of the resulting models. Authors show that GRU models with a hidden layer size of 4 units are able to accurately reproduce the system’s response to very different stimuli.

**Summary Of The Review:**

Overall its a weak accept. Authors should improve the automation in choosing appropriate stimuli for the training, validation and test sets as well as optimal parameter selection and perform a systematic analysis of compression possibilities of the learning-based models with error control.

---

### Decision · Program_Chairs · 2022-01-20

**Decision:**

Reject

**Comment:**

This paper explores the use of recurrent neural networks to model neural activity time-series data. The hope is that computationally demanding biophysical models of neural circuits could be replaced by RNNs when the goal is simply to capture the right input-output functions. The authors show that they can fit RNNs to the behaviour of a complex, biophysical model of the C elegans nervous system, and they explore the space of hyperparameter and network choices that lead to the best fits.

The reviews for this paper were borderline, with scores of 3, 6, and 8. On the positive side, the reviewers agreed that the paper is very effective in demonstrating that the input-output behaviour of the biophysical model of C elegans can be replicated by RNNs. But, on the negative side there were concerns about the limited nature of the empirical results, lack of details about the simulation, too much emphasis in describing well-known RNN architectures, and lack of systematic strategy for applying this technique in other systems. The rebuttals did not change the borderline scores.

Thus, this is an instance where the AC must be a bit more involved in the decision. After reading the paper and reviews, the AC felt that this work was not sufficiently general in its application. Ultimately, using artificial neural networks to fit neural data is common practice nowadays, so really, this paper serves as a proof-of-concept for replacing a complex biophysical model with a simpler RNN. But, given that RNNs are quite good at modelling sequence data, it's not terribly surprising that this works. Moreover, though the authors do a very careful search over network design decisions, they don't provide a systematic strategy for others to employ if they so wished. Also, the authors do not provide much insight into what the RNNs learn that might help us to better understand the modelled neural circuits. And most importantly, this only demonstrates the effectiveness for systems where we have biophysical models with well-established accuracy, which is not the case for most neural circuits. Given these considerations, a reject decision was reached.